# Hyperbolic matter in electrical circuits with tunable complex phases

Anffany Chen [1,2], Hauke Brand [3], Tobias Helbig [4], Tobias Hofmann [4], Stefan Imhof[3], Alexander Fritzsche[4,5], Tobias Kießling[3], Alexander Stegmaier[4], Lavi K. Upreti [4], Titus Neupert [6], Tomáš Bzdušek [6,7], Martin Greiter [4], Ronny Thomale [4] & Igor Boettcher [1,2] ✉

Curved spaces play a fundamental role in many areas of modern physics, from cosmological length scales to subatomic structures related to quantum information and quantum gravity. In tabletop experiments, negatively curved spaces can be simulated with hyperbolic lattices. Here we introduce and experimentally realize hyperbolic matter as a paradigm for topological states through topolectrical circuit networks relying on a complex-phase circuit element. The experiment is based on hyperbolic band theory that we confirm here in an unprecedented numerical survey of finite hyperbolic lattices. We implement hyperbolic graphene as an example of topologically nontrivial hyperbolic matter. Our work sets the stage to realize more complex forms of hyperbolic matter to challenge our established theories of physics in curved space, while the tunable complex-phase element developed here can be a key ingredient for future experimental simulation of various Hamiltonians with topological ground states.

Experimental Hamiltonian engineering and quantum simulation have become essential pillars of physics research, realizing artificial worlds in the laboratory with full control over tunable parameters and far-reaching applications from quantum many-body systems to high-energy physics and cosmology. Fundamental insights into the interplay of matter and curvature, for instance close to black hole event horizons or due to interparticle interactions[1–3], have been gained from the creation of synthetic curved spaces using photonic metamaterials[4,5]. The recent ground-breaking experimental implementation of hyperbolic lattices[6–8] in circuit quantum electrodynamics[9–11] and topolectrical circuits[12–15] constitutes another milestone in emulating curved space, separating the spatial manifold underlying the Hamiltonian entirely from its matter content to engineer broad classes of uncharted systems[16–19]. Conceptually, recent mathematical insights into hyperbolic lattices from algebraic geometry promise to inspire a fresh quantitative perspective onto curved space physics in general[20–22].

Hyperbolic lattices emulate particle dynamics that are equivalent to those in negatively curved space. They are two-dimensional lattices made from regular $p$-gons such that $q$ lines meet at each vertex, denoted $\{p, q\}$ for short, with $(p-2)(q-2) > 4$[6]. Such tessellations can only exist in the hyperbolic plane. In contrast, the Euclidean square and honeycomb lattices, $\{4, 4\}$ and $\{6, 3\}$, are characterized by $(p-2)(q-2) = 4$. Particle propagation on any of these lattices is described by the tight-binding Hamiltonian $\mathcal{H} = -J\sum_{\langle i,j\rangle}(c_i^\dagger c_j + c_j^\dagger c_i)$, with $c_i^\dagger$ the creation operator of particles at site $i$, $J$ the hopping amplitude, and the sum extending over all nearest neighbors.

In all previous experiments[6–8], hyperbolic lattices have been realized as finite planar graphs, or flakes, consisting of bulk sites with coordination number $q$ surrounded by boundary sites with

[1]Department of Physics, University of Alberta, Edmonton, AB T6G 2E1, Canada. [2]Theoretical Physics Institute, University of Alberta, Edmonton, AB T6G 2E1, Canada. [3]Physikalisches Institut, Universität Würzburg, 97074 Würzburg, Germany. [4]Institut für Theoretische Physik und Astrophysik, Universität Würzburg, 97074 Würzburg, Germany. [5]Institut für Physik, Universität Rostock, 18059 Rostock, Germany. [6]Department of Physics, University of Zurich, Winterthurerstrasse 190, 8057 Zurich, Switzerland. [7]Condensed Matter Theory Group, Paul Scherrer Institute, 5232 Villigen PSI, Switzerland. ✉e-mail: iboettch@ualberta.ca

coordination number $< q$. The ratio of bulk over boundary sites, as a fundamental property of hyperbolic space, is of order unity no matter how large the graph. Thus a large bulk system with negligible boundary, in contrast to the Euclidean case, can never be realized in a flake geometry. Instead, bulk observables on flakes always receive substantial contributions from excitations localized on the boundary. The isolation of bulk physics is thus crucial for understanding the unique properties of hyperbolic lattices.

In this work, we overcome the obstacle of the boundary and create a tabletop experiment that emulates genuine hyperbolic matter, which we define as particles propagating on an imagined infinite hyperbolic lattice, using topolectrical circuits with tunable complex-phase elements. This original method creates an effectively infinite hyperbolic space without the typical extensive holographic boundary −our system consists of pure bulk matter instead. The setup builds on hyperbolic band theory, which implies that momentum space of two-dimensional hyperbolic matter is four-, six- or higher-dimensional, as we confirm here numerically for finite hyperbolic lattices with both open and periodic boundary conditions. We introduce and implement hyperbolic graphene and discuss its topological properties and Floquet physics. Our work paves the way for theoretical studies of more complex hyperbolic matter systems and their experimental realization.

## Results

### Infinite hyperbolic lattices as unit-cell circuits

The key to simulating infinite lattices is to focus on the wave functions of particles on the lattice. In Euclidean space, Bloch's theorem states that under the action of the two translations generating the Bravais lattice, denoted $T_1$ and $T_2$, a wave function $\psi_{\mathbf{k}}(z_i)$ transforms as

$$\psi_{\mathbf{k}}(T_\mu^{-1} z_i) = e^{ik_\mu} \psi_{\mathbf{k}}(z_i). \tag{1}$$

Here $z_i$ is any site on the lattice, $\mathbf{k} = (k_1, k_2)$ is the crystal momentum with $\mu = 1, 2$, and $e^{ik_\mu}$ is the complex Bloch phase factor. In crystallography, we split the lattice into its Bravais lattice and a reference unit cell of $N$ sites with coordinates $z_n, n \in \{1, ..., N\}$. The full wave function is obtained from the values in the unit cell by successive application of Eq. (1). Furthermore, the energy bands on the lattice in the tight-binding limit, $\varepsilon_n(\mathbf{k})$, are the eigenvalues of the $N \times N$ Bloch-wave Hamiltonian matrix $H(\mathbf{k})$. In the latter, the matrix entry at position $(n, n')$ is the sum of all Bloch phases for hopping between neighboring sites $z_n$ and $z_{n'}$ after endowing the unit cell with periodic boundaries. (See Methods for an explicit construction algorithm of $H(\mathbf{k})$.) The approach is visualized in Fig. 1a and b for the $\{6, 3\}$ honeycomb lattice with $N = 2$ unit cell sites. The

associated $2 \times 2$ Bloch-wave Hamiltonian is

$$H_{\{6,3\}}(\mathbf{k}) = -J \begin{pmatrix} 0 & 1 + e^{ik_1} + e^{ik_2} \\ 1 + e^{-ik_1} + e^{-ik_2} & 0 \end{pmatrix}, \tag{2}$$

with eigenvalues $\varepsilon_\pm(\mathbf{k}) = \pm J|1 + e^{ik_1} + e^{ik_2}|$. This models the band structure of graphene in the non-interacting limit[23,24].

Recent theoretical insights into hyperbolic band theory (HBT) and non-Euclidean crystallography revealed that this construction also applies to hyperbolic lattices, as many of them split into Bravais lattices and unit cells[20,25]. There are two crucial differences between two-dimensional Euclidean and hyperbolic lattices. First, the number of hyperbolic translation generators is larger than two, denoted $T_1, ..., T_{2g}$, with integer $g > 1$. Second, hyperbolic translations do not commute, $T_\mu T_{\mu'} \neq T_{\mu'} T_\mu$. Nonetheless, Bloch waves transforming as in Eq. (1) can be eigenfunctions of the Hamiltonian $\mathcal{H}$ on the infinite lattice. These solutions are labelled by $2g$ momentum components $\mathbf{k} = (k_1, ..., k_{2g})$ from a higher-dimensional momentum space. The dimension of momentum space is defined as the number of generators of the Bravais lattice. The associated energy bands $\varepsilon_n(\mathbf{k})$ are computed from the Bloch-wave Hamiltonian $H(\mathbf{k})$ in the same manner as described above.

We are lead to the important conclusion that Bloch-wave Hamiltonians $H(\mathbf{k})$ of both Euclidean and hyperbolic $\{p, q\}$ lattices are equivalent to unit-cell circuits with $N$ vertices of coordination number $q$. Bloch phases $e^{i\phi(\mathbf{k})}$ are imprinted along certain edges in one direction and $e^{-i\phi(\mathbf{k})}$ in the opposite direction, see Fig. 1d. Examples are visualized in Fig. 1c, e, f. The infinite extent of space is implemented through distinct momenta $\mathbf{k}$. Due to the non-commutative nature of hyperbolic translations, other eigenfunctions of $\mathcal{H}$ in higher-dimensional representations exist besides Bloch waves. They are labelled by an abstract $\mathbf{k}$, where $\psi_{\mathbf{k}}$ in Eq. (1) has $d > 1$ components and Bloch phases $e^{i\phi(\mathbf{k})}$ are $d \times d$ unitary matrices. Presently very little is known about these states[21,22], but we demonstrate in this work that ordinary Bloch waves capture large parts of the spectrum on hyperbolic lattices.

### Tunable complex phases in electrical networks

Topolectrical circuit networks are an auspicious experimental platform for implementing unit-cell circuits. In topolectrics, tight-binding Hamiltonians defined on finite lattices are realized by the graph Laplacian of electrical networks[12–14]. Wave functions and their corresponding energies can be measured efficiently at every lattice site. While the real-valued edges in unit-cell circuits can be implemented using existing technology[14], we had to develop a tunable complex-phase element to imprint the non-vanishing Bloch phases $e^{i\phi(\mathbf{k})}$. Importantly, while circuit elements existed before that realize a fixed

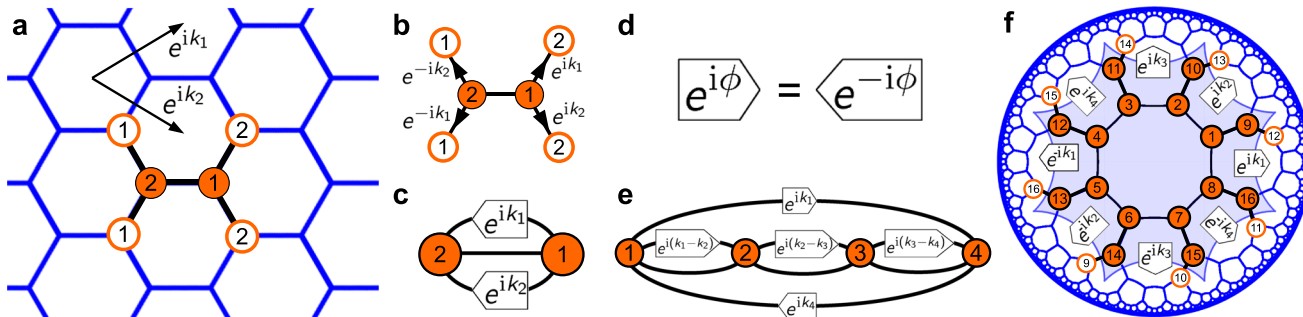

**Fig. 1 | Unit-cell circuits. a** Euclidean $\{6, 3\}$ honeycomb lattice with two sites in the unit cell (full orange circles). Each site has 3 neighbors, some of them in adjacent unit cells (empty orange circles). **b** The wave function of particles hopping between unit cells picks up a complex Bloch phase, see Eq. (1). **c** The associated unit-cell circuit diagram encodes the Bloch-wave Hamiltonian $H(\mathbf{k})$, Eq. (2), and the energy bands. Momentum $\mathbf{k} = (k_1, k_2)$ is an external parameter. **d** In topolectrical circuits, a

complex-phase element imprints tunable Bloch phases along edges connecting neighboring sites. The circuit element is directed, with $e^{i\phi}$ imprinted in one direction, and $e^{-i\phi}$ in the other. This leads to Hermitian matrices $H(\mathbf{k})$. Unit-cell circuits for the $\{8, 4\}$ (**e**) and $\{8, 3\}$ (**f**) hyperbolic lattices. The Bravais lattice is the $\{8, 8\}$ lattice in either case, with 4 and 16 sites in the unit cell, respectively. In these lattices, Bloch waves carry a four-dimensional momentum $\mathbf{k} = (k_1, k_2, k_3, k_4)$.

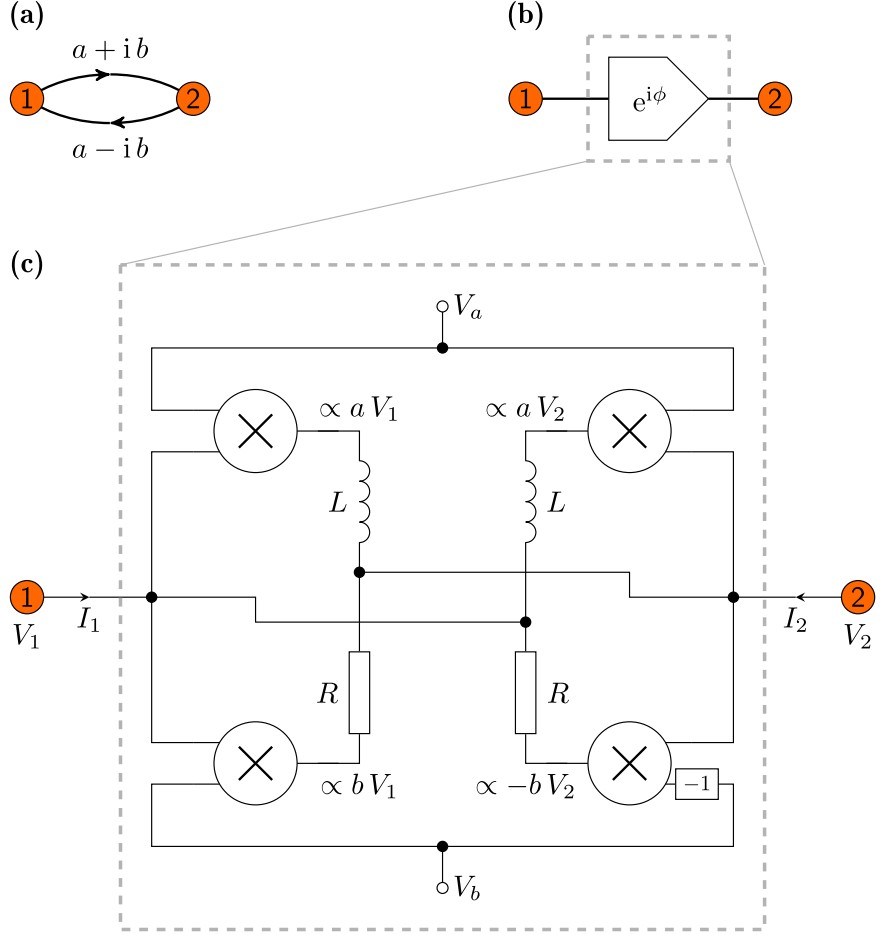

**Fig. 2 | Tunable complex-phase element. a** Hermitian hopping term $a \pm ib$ which is to be implemented between two nodes 1 and 2 in an electric circuit. **b** Symbol for the circuit element corresponding to the hopping term with $e^{i\phi} \propto a + ib$. The impedance representation is given by Eq. (3) with $a = \cos(\phi)/(i\omega L)$ and $b = \sin(\phi)/(i\omega L)$. **c** Implementation of the circuit element using four analog multipliers (represented by the circles with a cross symbol). We choose $R = \omega L$. The voltages $V_a$ and $V_b$ tune the phase $\phi = \arctan V_b / V_a$. This circuit implements the complex coupling from node 1 to 2 with phase $e^{i\phi}$ as well as the back-direction from 2 to 1 with phase $e^{-i\phi}$.

complex phase $e^{i\phi}$ along an edge[8,26], changing the value of $e^{i\phi}$ required to dismantle the circuit and modify the element. In contrast, the phase $e^{i\phi}$ of the element constructed here can be tuned by varying external voltages applied to the circuit. In the future, this highly versatile circuit element can be applied in multifold physical settings beyond realizing hyperbolic matter, including synthetic dimensions and synthetic magnetic flux threading.

The schematic structure of the circuit element is shown in Fig. 2. It contains four analog multipliers, the impedance of which is chosen to be either resistive (for the bottom two multipliers) or inductive (for the top two multipliers). As detailed in Methods, their outputs are connected in such a way that the circuit Laplacian of the element reads

$$\begin{pmatrix} I_1 \\ I_2 \end{pmatrix} = \frac{1}{i\omega L} \begin{pmatrix} 1+i & e^{-i\phi} \\ e^{i\phi} & 1+i \end{pmatrix} \begin{pmatrix} V_1 \\ V_2 \end{pmatrix}, \qquad (3)$$

where $I_1$ and $I_2$ are the currents flowing into the circuit from the points at potentials $V_1$ and $V_2$, respectively. The diagonal entries merely result in a constant shift of the admittance spectrum. The off-diagonal entries are controlled by external voltages $V_a$ and $V_b$ according to $V_b/V_a = \tan\phi$, so $\phi$ is tunable, with resolution limited only by the resolution of the sources that provide those voltages. Equation (3) therefore realizes a Bloch-wave term with $\phi = \phi(\mathbf{k})$.

## Validity of Bloch-wave assumption

Unit-cell circuits of hyperbolic lattices only capture the Bloch-wave eigenstates of the hyperbolic translation group. To test how well this approximates the full energy spectrum on infinite lattices resulting from both Bloch waves and higher-dimensional representations, we compare the predictions of HBT for the density of states (DOS) to results obtained from exact diagonalization on finite $\{p, q\}$ lattices with up to several thousand vertices and either open boundary conditions (flakes) or periodic boundary conditions (regular maps). In the case of flake geometries[6,18], the boundary effect on the DOS can be partly eliminated by considering the bulk-DOS[17,27,28], defined as the sum of local DOS over all bulk sites (see Methods). To implement periodic boundary conditions, we utilize finite graphs known as regular maps[29–32], which are $\{p, q\}$ tessellations of closed hyperbolic surfaces with constant coordination number $q$ that preserve all local point-group symmetries of the lattice.

For the comparison, we consider lattices of type $\{7, 3\}$, $\{8, 3\}$, $\{8, 4\}$, $\{10, 3\}$, and $\{10, 5\}$. This selection is motivated by the possibility to split these lattices into unit cells and Bravais lattices, and hence to construct the Bloch-wave Hamiltonian $H_{\{p,q\}}(\mathbf{k})$[25]. Our extensive numerical analysis, presented in Supplementary Info. Secs. I–III, shows that both bulk-DOS on large flakes and DOS on large regular maps converge to universal functions determined by $p$ and $q$. We find that HBT yields accurate predictions of the DOS for lattices $\{7, 3\}$, $\{8, 3\}$, and $\{10, 3\}$, see Fig. 3.

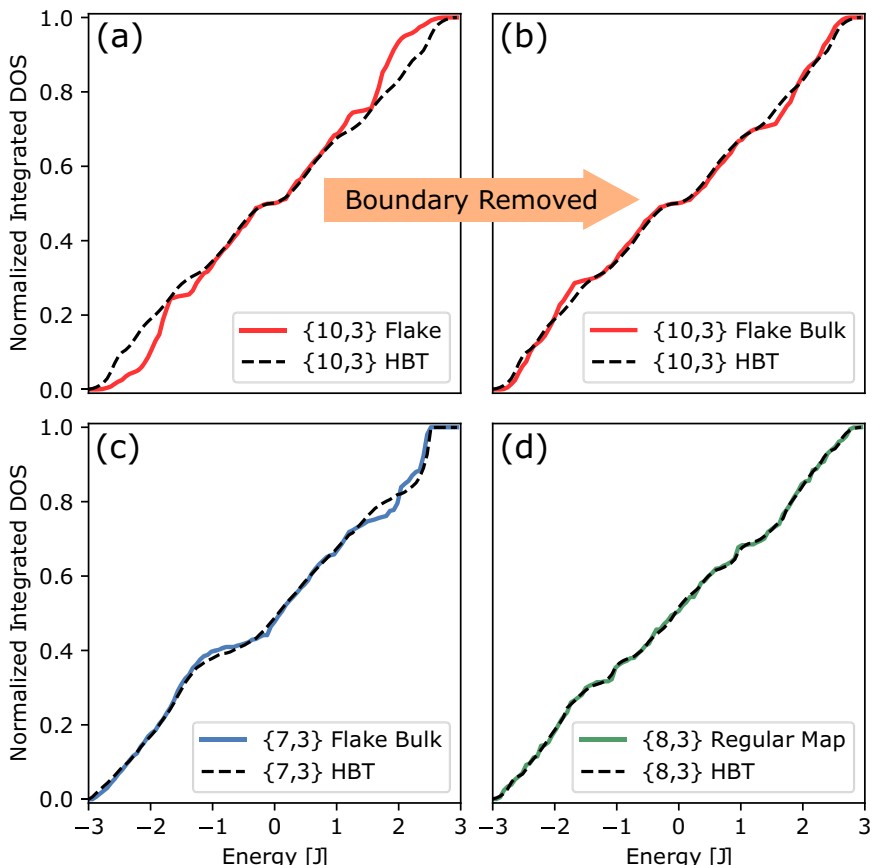

**Fig. 3 | Density of states.** Integrated DOS computed from finite {$p$, 3} lattices vs. predictions from hyperbolic band theory (HBT) realized in unit-cell circuits. **a** DOS of a {10, 3} flake with 2880 sites. **b** Bulk-DOS of the same lattice as in **a**. With the boundary contribution removed, it agrees well with band theory. **c** Bulk-DOS of a {7, 3} flake with 847 sites vs. band theory. **d** The averaged DOS of five {8, 3} regular maps (each with ~2000 sites) reveals excellent agreement with band theory.

Generally, the agreement between HBT and regular maps is better than for flake geometries, likely since no subtraction of boundary states is needed. For some regular maps, called Abelian clusters[21], HBT is exact and all single-particle energies on the graph read $\varepsilon_n(\mathbf{k}_i)$ with certain quantized momenta $\mathbf{k}_i$. We explore their connection to higher-dimensional Euclidean lattices in Supplementary Info. Sec. S III.

For the {8, 4} and {10, 5} lattices, we find that the bulk-DOS on hyperbolic flakes deviates more significantly from the predictions of HBT. This may originate from (i) the omission of higher-dimensional representations or (ii) enhanced residual boundary contributions to the approximate bulk-DOS. The latter is due to the larger boundary ratio for {8, 4} and {10, 5} lattices (see Supplementary Info. Table S2). Despite the deviation, studying Bloch waves on these lattices, and their contribution to band structure or response functions, is an integral part of understanding transport in these hyperbolic lattices. Investigating the extent to which higher-dimensional representations mix with Bloch waves (selection rules) will shed light on their role in many-body or interacting hyperbolic matter in the future.

Note that the unit-cell circuits can be adapted to simulate non-Abelian Bloch states. One such option is to use a specific irreducible representation as an ansatz for constructing the corresponding non-Abelian eigenstates[22,33]. If the representation is $d$-dimensional, then the non-Abelian Bloch Hamiltonian can be emulated by building a circuit with $d$ degrees of freedom on each node, giving a total of $Nd$ nodes in the unit cell circuit.

## Hyperbolic graphene

We define hyperbolic graphene as the collection of Bloch waves on the {10, 5} lattice, realized by its unit-cell circuit depicted in Fig. 4a.

The {10, 5} lattice has two sites in its unit cell and four independent translation generators, resulting in the Bloch-wave Hamiltonian

$$H_{\{10,5\}}(\mathbf{k}) = -J \begin{pmatrix} 0 & h(\mathbf{k}) \\ h(\mathbf{k})^* & 0 \end{pmatrix}, \tag{4}$$

$$h(\mathbf{k}) = 1 + e^{ik_1} + e^{ik_2} + e^{ik_3} + e^{ik_4}, \tag{5}$$

with crystal momentum $\mathbf{k} = (k_1, k_2, k_3, k_4)$ (see Supplementary Info. Sec. S I for explicit construction). The two energy bands read $\varepsilon_{\pm}(\mathbf{k}) = \pm J|h(\mathbf{k})|$. Hyperbolic graphene mirrors many of the enticing properties of graphene on the {6, 3} lattice (henceforth assumed non-interacting with only nearest-neighbor hopping). Both systems belong to a larger family of {2(2$g$+1), 2$g$+1} Bravais lattices with two-site unit cells and 2$g$ translation generators[25]. Restricting the sum in Eq. (5) to two complex phases, we obtain Eq. (2). In fact, hyperbolic graphene contains infinitely many copies of graphene through setting $k_3 = k_4 + \pi$ in $h(\mathbf{k})$.

The most striking resemblance between hyperbolic graphene and its Euclidean counterpart is the emergence of Dirac particles at the band crossing points. These form a nodal surface $\mathcal{S}$ in momentum space, determined by the condition $h(\mathbf{k}) = 0$. This is a complex equation and thus results in a manifold of real co-dimension two. Whereas this implies isolated Dirac points in graphene, the nodal surface of Dirac excitations in hyperbolic graphene is two-dimensional because momentum space is four-dimensional, see Fig. 4b. The associated Dirac Hamiltonian is derived in Supplementary Info. Sec. S IV. At each Dirac point $\mathbf{k}_0 \in \mathcal{S}$, momentum space splits into a tangential and

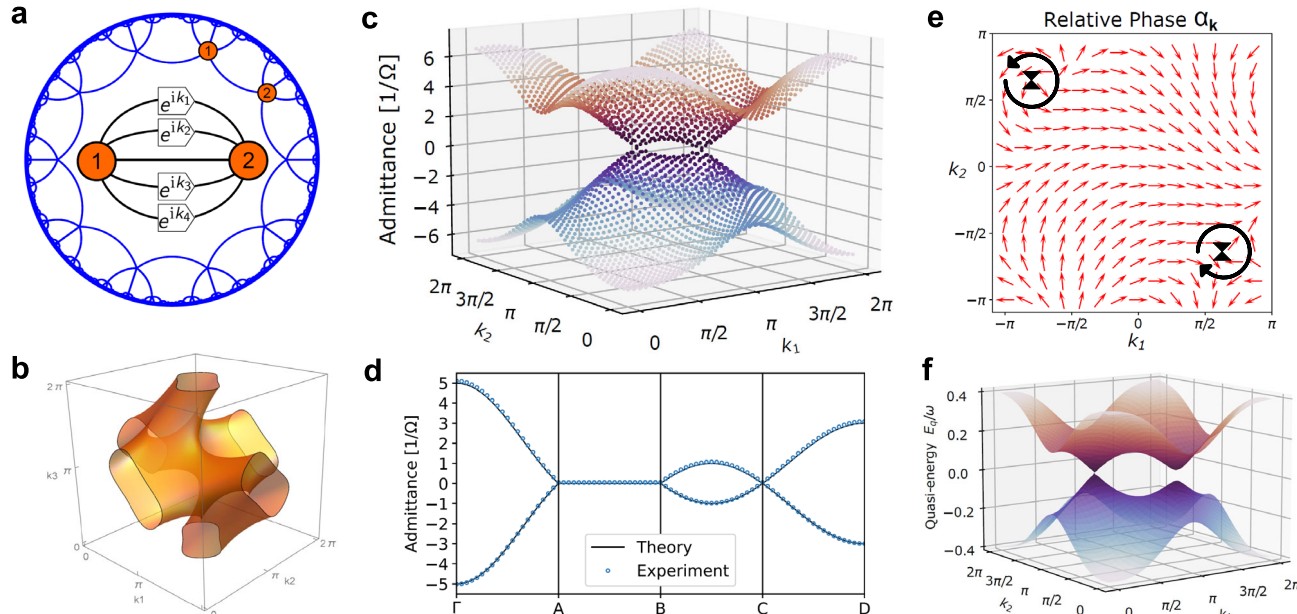

**Fig. 4 | Hyperbolic graphene.** This collection of Bloch waves on the hyperbolic {10, 5} lattice features many properties of its Euclidean counterpart, but with a hyperbolic twist. It is a topological semimetal with four-dimensional momentum space and crystal momentum $\mathbf{k} = (k_1, k_2, k_3, k_4)$. **a** Two sites in the unit cell with five nearest neighbors comprise the unit-cell circuit that we realize experimentally with topolectrics. **b** Two-dimensional nodal surface $\mathcal{S}$ of gapless Dirac excitations with energy $|h(\mathbf{k})| = 0$, projected onto the $(k_1, k_2, k_3)$-hyperplane. **c** Experimentally measured Dirac cones in the plane $\mathbf{k} = (k_1, k_2, 2\pi/3, 0)$ as a function of $k_1$ and $k_2$. **d** Experimentally measured spectrum along path through $\mathbf{k}$-space. The labels Γ, A, B, C, D correspond to the Brillouin zone points $(0, 0, 0, 0)$, $(2\pi/3, -2\pi/3, 0, \pi)$, $(-2\pi/3, 0, 2\pi/3, \pi)$, $(2\pi/3, 0, -2\pi/3, \pi)$, $(\pi, \pi, \pi, \pi)$, respectively. Experimental errors in c) and d) are smaller than plotted data points. The overall energy scale is matched to the theoretical model by a rescaling of the circuit Laplacian. **e** In the momentum plane $\mathbf{k} = (k_1, k_2, 0, \pi)$, the Berry phase computed along a closed loop surrounding each node is $\pi$. This can be seen from the vortex-antivortex-pair formed by the phase $\alpha_{\mathbf{k}}$ of the eigenstates. **f** Periodically driven hyperbolic graphene in the high-frequency, low-amplitude regime features non-uniform gap opening over the nodal region. We show theoretical predictions for the quasi-energy spectrum driven according to $k_\mu(t) = k_\mu^{(0)} + 0.8 \sin(6t + \mu\pi/2)$ in the momentum plane $\mathbf{k}^{(0)} = (k_1, k_2, 2\pi/3, -\pi/3)$.

normal plane. Within the latter, a $\pi$ Berry phase can be computed along a loop enclosing the Dirac point, protected by the product of time-reversal and inversion symmetries[34,35]. Therefore, hyperbolic graphene is a synthetic topological semimetal and a platform to study topological states of matter. Its momentum-space topology is the natural four-dimensional analogue of two-dimensional graphene and three-dimensional nodal-line semimetals[36].

We experimentally realized the unit-cell circuit for hyperbolic graphene in topolectrics with four tunable complex-phase elements. The circuit represents the Hamiltonian $H_{\{10, 5\}}(\mathbf{k})$ at any desired point in the four-dimensional Brillouin zone. We measured the band structure in the two-dimensional plane defined by $\mathbf{k} = (k_1, k_2, 2\pi/3, 0)$ for varying $k_1$, $k_2$, which contains exactly two Dirac points, see Fig. 4c. We also obtained the accompanying eigenstates. In Fig. 4d, we measured the band structure along lines connecting representative points in the Brillouin zone. This further highlights both the tunability of the experimental setup and the extended band-touching region of the model in momentum space, in contrast to the isolated nodal points in Euclidean graphene.

To visualize the nontrivial topology of hyperbolic graphene, we write the eigenstates as $|\psi_{\mathbf{k}}^{\pm}\rangle = (1, \pm e^{i\alpha_{\mathbf{k}}})$. The phase $\alpha_{\mathbf{k}}$ changes by $2\pi$ upon encircling a Dirac node in the normal plane, creating a momentum-space vortex, and $|\psi_{\mathbf{k}}^{\pm}\rangle$ picks up a Berry phase of $\pi$ (see Methods). We numerically compute the lower-energy eigenstates $|\psi_{\mathbf{k}}^{-}\rangle$ in the two-dimensional plane defined by $\mathbf{k} = (k_1, k_2, 0, \pi)$ and observe a vortex-antivortex pair, see Fig. 4e. While the nontrivial Berry phase in graphene implies zero-energy boundary modes, the bulk-boundary correspondence in hyperbolic graphene is complicated by the mismatch of position- and momentum-space dimensions, see Supplementary Info. Sec. S V.

By periodic tuning of the complex-phase elements, it is also possible to imitate the effect of irradiation of charged carriers in hyperbolic lattices. In this context, recall that graphene irradiated by circularly polarized light, modelled by electric field $\mathbf{E}(t) = \partial \mathbf{a}(t)/\partial t$ and vector potential $\mathbf{a}(t) = a_0(\sin(\omega t), \cos(\omega t))$, where $\omega$ is the frequency of light, realizes a Floquet system with topologically nontrivial band gaps[37,38]. In the unit-cell picture, this can be simulated by a fast periodic driving of the external momentum on time scales much shorter than the measurement time, parametrized as $k_\mu(t) = k_\mu - A\sin(\omega t + \varphi_\mu)$, with driving amplitude $A = ea_0$ and phase shift $\varphi_\mu$. We theoretically demonstrate that hyperbolic graphene with such $k_\mu(t)$ exhibits characteristic gap opening in the Floquet regime, though the gap size varies over the nodal region in contrast to graphene (see Fig. 4f). Notably, part of the nodal region remains approximately gapless within the energy resolution of the experiment (see Methods), bearing potential to study exotic transport phenomena far from equilibrium.

## Discussion

This work paves the way for several highly exciting future research directions in both experimental and theoretical condensed matter physics. Experimentally, the tunable complex-phase element developed here can be utilized in topolectrical networks to simulate Hamiltonians with topological ground states, such as the recently discovered hyperbolic topological band insulators[28,39] or hyperbolic Hofstadter butterfly models[17,32]. In particular, local probes in electric circuits provide access to the complete characterization of the Bloch eigenstates, giving the necessary input to compute any topological invariant. We have shown how synthetic extra dimensions can be emulated efficiently through tunable complex phase elements, which may be used in conjunction with ordinary one- or two-dimensional

lattices to create effectively higher-dimensional Euclidean or hyperbolic models. Electric circuits also admit measurements of the time-resolved evolution of states, thus giving access to various non-equilibrium phenomena beyond the Floquet experiment discussed in the text. Additionally, together with nonlinear, non-Hermitian or active circuit elements[40–42], interaction effects beyond the single-particle picture can be captured in these models, allowing for experimental engineering of a wide range of Hamiltonians.

Theoretically, hyperbolic matter constitutes a paradigm for topological states of matter with many surprising and unique physical features, which are hinted at by the original energetic and topological properties of hyperbolic graphene with Dirac particles in four-dimensional momentum space. By joining multiple unit-cell circuits, multi-layer settings can be emulated: for instance, using two real-valued connections to join the same sublattice sites of hyperbolic graphene realizes AA-stacked bilayer hyperbolic graphene. Such studies will shed more light on the subtle interplay between lattice structure and energy bands, a topic that recently came into the focus of many researchers with the fabrication of moiré materials[43]. The mismatch of position- and momentum-space dimensions requires to re-evaluate many properties of Dirac particles in the context of hyperbolic graphene such as the bulk-boundary correspondence discussed earlier, or Klein tunneling and Zitterbewegung, which have been observed in one-dimensional Euclidean condensed matter systems[44–46] and discussed for graphene[23,47].

## Methods
### Bloch-wave Hamiltonian matrix
One can construct the Bloch-wave Hamiltonian matrix $H(\mathbf{k})$ of a $\{p, q\}$ hyperbolic lattice if it can be decomposed into a $\{p_B, q_B\}$ Bravais lattice with a unit cell of $N$ sites, denoted $\{z_n\}_{n=1,...,N}$. The matrix is constructed as follows. (i) Initially set all entries of the matrix $H(\mathbf{k})$ to zero. (ii) For each unit cell site $z_n$, determine the $q$ neighboring sites $z_i$. (iii) For each neighbor $z_i$, determine the translation $T^{(i)}$ such that $z_i = T^{(i)}z_m$ for some $z_m$ in the unit cell. (iv) If $T^{(i)} = 1$, add 1 to $H_{nm}(\mathbf{k})$, otherwise add the Bloch phase $e^{i\phi(\mathbf{k})}$ that is picked up when going from $z_n$ to $z_i$. (v) Multiply the matrix by $-J$. The detailed procedure for the lattices considered in this work is documented in Supplementary Info Sec. S I. A list of known hyperbolic lattices with their corresponding Bravais lattices and unit cells is given in ref. [25].

### Hamiltonian of real-space hyperbolic lattices
The Hamiltonians of hyperbolic lattices with open boundary conditions (flakes) were generated by the shell-construction method used in refs. [6,25]. One obtains the Poincaré coordinates of the lattice sites and the adjacency matrix $\mathcal{A}$, where $\mathcal{A}_{ij}$ is 1 if sites $i$ and $j$ are nearest neighbours and 0 otherwise. The tight-binding Hamiltonian in first-quantized form is then $\mathcal{H} = -J\mathcal{A}$, where $J$ is the hopping amplitude. The adjacency matrices of hyperbolic lattices with periodic boundary condition (regular maps) were identified from mathematical literature[30] and are listed in Supplementary Info Table S3. A larger set of hyperbolic regular maps has been identified in ref. [32].

### Bulk-DOS of hyperbolic flakes
To effectively remove the boundary contribution to the total DOS of a hyperbolic flake, we define the bulk-DOS as the sum of the local DOS over all bulk sites through

$$\rho_{\text{bulk}}(\epsilon) = \sum_{z \in \Lambda_{\text{bulk}}} \left( \sum_{n \in \mathcal{N}_\epsilon} |\psi_n(z)|^2 \right). \tag{6}$$

Here, $\Lambda_{\text{bulk}}$ is the set of lattice sites with coordination number equal to $q$ and $\mathcal{N}_\epsilon$ is the set of eigenstates with energies between $\epsilon$ and

$\epsilon + \delta\epsilon$. In the DOS comparison, we use the normalized integrated DOS (or spectral staircase function)

$$P_{\text{bulk}}(E) = \frac{\int_{-q}^{E} d\epsilon \, \rho_{\text{bulk}}(\epsilon)}{\int_{-q}^{q} d\epsilon \, \rho_{\text{bulk}}(\epsilon)}. \tag{7}$$

This quantity is approximately independent of system size (number of shells), see Supplementary Info. Fig. S2. Note that the energy spectrum of a $\{p, q\}$ lattice is in the range $[-q, q]$.

### Dirac nodal region of hyperbolic graphene
The Bloch-wave Hamiltonian of hyperbolic graphene can be written as

$$H_{\{10,5\}}(\mathbf{k}) = d_x(\mathbf{k})\sigma_x + d_y(\mathbf{k})\sigma_y, \tag{8}$$

where $d_x(\mathbf{k}) = -1 - \sum_{\mu=1}^{4} \cos(k_\mu)$ and $d_y(\mathbf{k}) = -\sum_{\mu=1}^{4} \sin(k_\mu)$ with hopping amplitude $J$ set to 1. The energy bands are $\varepsilon_\pm(\mathbf{k}) = \pm\sqrt{d_x(\mathbf{k})^2 + d_y(\mathbf{k})^2}$, so the band-touching region is determined by the two equations $d_x(\mathbf{k}) = 0$ and $d_y(\mathbf{k}) = 0$. With four $\mathbf{k}$-components, these two equations define the two-dimensional nodal surface $\mathcal{S}$ visualized in Fig. 4(b). Near every node $\mathbf{Q} \in \mathcal{S}$, $H_{\{10,5\}}(\mathbf{k})$ is approximated by the Dirac Hamiltonian

$$h_{\text{eff}}^{\mathbf{Q}}(\mathbf{q}) = \sigma_x \mathbf{q} \cdot \mathbf{u}(\mathbf{Q}) - \sigma_y \mathbf{q} \cdot \mathbf{v}(\mathbf{Q}) + \mathcal{O}(q^2), \tag{9}$$

where $\mathbf{u}(\mathbf{Q}) = \sum_{\mu=1}^{4} \sin(Q_\mu)\mathbf{e}_\mu$ and $\mathbf{v}(\mathbf{Q}) = \sum_{\mu=1}^{4} \cos(Q_\mu)\mathbf{e}_\mu$. Here $\mathbf{e}_\mu$ is the unit vector in the direction of $k_\mu$. For the detailed derivation, see Supplementary Info. Sec. S IV.

### Berry phase in hyperbolic graphene
We write Eq. (8) as

$$H_{\{10,5\}}(\mathbf{k}) = \begin{pmatrix} 0 & r_\mathbf{k}e^{-i\alpha_\mathbf{k}} \\ r_\mathbf{k}e^{i\alpha_\mathbf{k}} & 0 \end{pmatrix} \tag{10}$$

with $r_\mathbf{k} = \sqrt{d_x(\mathbf{k})^2 + d_y(\mathbf{k})^2}$ and $\alpha_\mathbf{k} = \arctan(d_y(\mathbf{k})/d_x(\mathbf{k}))$. The eigenstates are $|\psi_\mathbf{k}^\pm\rangle = (1, \pm e^{i\alpha_\mathbf{k}})$. The relative phase $\alpha_\mathbf{k}$ undergoes a $2\pi$ rotation around any given node $\mathbf{Q} \in \mathcal{S}$, implying a $\pi$ Berry phase. One can verify this numerically by taking a chain of momenta $\{\mathbf{k}_1, \mathbf{k}_2, ..., \mathbf{k}_n\}$ on the closed loop $\mathbf{k}(s) = \mathbf{Q} + \mathbf{u}(\mathbf{Q})\cos(s) + \mathbf{v}(\mathbf{Q})\sin(s)$, $s \in [0, 2\pi]$, and then using the lower-energy state to compute the Berry phase, given by $\gamma = \text{Im} \ln (\langle\psi_{\mathbf{k}_1}^-|\psi_{\mathbf{k}_2}^-\rangle\langle\psi_{\mathbf{k}_2}^-|\psi_{\mathbf{k}_3}^-\rangle \cdots \langle\psi_{\mathbf{k}_n}^-|\psi_{\mathbf{k}_1}^-\rangle)$ in the discrete formulation[48].

### Floquet band gaps in hyperbolic graphene
With tunable complex-phase elements, it is possible to drive individual momentum components of hyperbolic graphene periodically, realizing the time-dependent Hamiltonian

$$H_{\{10,5\}}(\mathbf{k},t) = -J \begin{pmatrix} 0 & 1 + \sum_{\mu=1}^{4} e^{i(k_\mu - A\sin(\omega t + \varphi_\mu))} \\ \text{c.c.} & 0 \end{pmatrix}, \tag{11}$$

where $A$ is the driving amplitude, $\omega$ is the frequency, and $\varphi_\mu$ are offsets in the periodic drive. Applying Floquet theory[49] and degenerate perturbation theory[50] near a Dirac node $\mathbf{k} \in \mathcal{S}$, we determine the effective Hamiltonian in the limit $A \ll 1$ and $\omega \gg J$, to order $\mathcal{O}(A^4)$, to be

$$H_{\text{eff}}(\mathbf{k}) = -J \begin{pmatrix} 0 & 1 + \mathcal{J}_0(A)\sum_{\mu=1}^{4} e^{ik_\mu} \\ \text{c.c.} & 0 \end{pmatrix} + \Delta(\mathbf{k})\sigma_z. \tag{12}$$

Here $\mathcal{J}_0(A)$ is the zeroth Bessel function of the first kind and

$$\Delta(\mathbf{k}) = \frac{J^2 A^2}{2\omega} \sum_{\mu=1} \sum_{\nu=1:\nu\neq\mu} \sin(k_\mu - k_\nu) \sin(\varphi_\mu - \varphi_\nu). \quad (13)$$

The factor of $\mathcal{J}_0(A)$ in the first term of Eq. (12) slightly shifts the location of the node while the second term opens up a $\mathbf{k}$-dependent gap $\Delta(\mathbf{k})$. Clearly, if the phases $\varphi_\mu$ are identical, $\Delta(\mathbf{k})$ is trivial. For a generic set of phases $\varphi_\mu$, however, there exists a one-dimensional subspace of $\mathcal{S}$ where $\Delta(\mathbf{k}) = 0$, implying that the nodes remain gapless up to $\mathcal{O}(A^4)$. See Supplementary Info. Sec. VI for a more detailed derivation and discussion of the Floquet equations relevant for this work.

### Tunable complex-phase element

In the following we specify the components used in the circuit shown in Fig. 2 and derive Eq. (3). More technical details together with more detailed illustrations are given in Supplementary Info. Secs. VII and VIII.

The complex-phase element as shown in Fig. 2 features four AD633 analog multipliers by Analog Devices Inc. The transfer function of these multipliers is given by $W = \frac{(X_1 - X_2)\cdot(Y_1 - Y_2)}{10\,\mathrm{V}} + Z$, where $W$ is the output, $X_1, X_2, Y_1, Y_2$ are the inputs (with $X_2$ and $Y_2$ inverted), and $Z$ is an additional input. Note that 10 V is the reference voltage for the analog multipliers. The other components include the SRR7045-471M inductors, with a nominal inductance of $470\,\mu H$ at 1kHz, which were selected to minimize variance in the inductance. To achieve tunability of the resistance value, the resistors connected to the bottom multipliers are the $50\,\Omega$ PTF6550R000BYBF resistor and the $50\,\Omega$ Bourns 3296W500 potentiometer in series.

To derive the circuit Laplacian of the complex-phase element as defined in Eq. (3), we consider the voltage drops over individual inductors and resistors in Fig. 2. First let us consider the pair on the left. The voltage drops are determined by the output voltages of the left multipliers and therefore equal to $\frac{V_a V_1}{10\,\mathrm{V}} - V_2$ and $\frac{V_b V_1}{10\,\mathrm{V}} - V_2$ for the inductor and resistor respectively. The current $I_2$ is then the negated sum of these voltage drops, each multiplied by the respective admittance: $I_2 = -\left(\frac{1}{i\omega L}\left(\frac{V_a V_1}{10\,\mathrm{V}} - V_2\right) + \frac{1}{R}\left(\frac{V_b V_1}{10\,\mathrm{V}} - V_2\right)\right)$. The relationship between the current $I_1$ and the applied voltages can be derived in the same fashion, yielding $I_1 = -\left(\frac{1}{i\omega L}\left(\frac{V_a V_2}{10\,\mathrm{V}} - V_1\right) + \frac{1}{R}\left(\frac{-V_b V_2}{10\,\mathrm{V}} - V_1\right)\right)$. One then obtains Eq. (3) by further choosing $R = \omega L$ and applying voltage signals of 10 V $\sin(\phi)$ and 10 V $\cos(\phi)$ to $V_a$ and $V_b$ respectively.

## Data availability

All the data (both experimental data and data obtained numerically) used to arrive at the conclusions presented in this work are publicly available in the following data repository: https://doi.org/10.5683/SP3/EG9931.

## Code availability

All the Wolfram Language code used to generate and/or analyze the data and arrive at the conclusions presented in this work is publicly available in the form of annotated Mathematica notebooks in the following data repository: https://doi.org/10.5683/SP3/EG9931.

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

## Acknowledgements
We thank A. Fahimniya, A. Gorshkov, A. Kollár, P. Lenggenhager, and J. Maciejko for inspiring discussions. A.C. and I.B. acknowledge support from the University of Alberta startup fund UOFAB Startup Boettcher. I.B. acknowledges funding from the Natural Sciences and Engineering Research Council of Canada (NSERC) Discovery Grants RGPIN-2021-02534 and DGECR2021-00043. The work in Würzburg is funded by the Deutsche Forschungsgemeinschaft (DFG, German Research Foundation) through Project-ID 258499086 - SFB 1170 and through the Würzburg-Dresden Cluster of Excellence on Complexity and Topology in Quantum Matter – *ct.qmat* Project-ID 39085490 - EXC 2147. T.He. was supported by a Ph.D. scholarship of the German Academic Scholarship Foundation. T.N. acknowledges funding from the European Research Council (ERC) under the European Union's Horizon 2020 research and innovation programm (ERC-StG-Neupert-757867-PARATOP). T.B. was supported by the Ambizione grant No. 185806 by the Swiss National Science Foundation.

## Author contributions
I.B. and R.T. initiated the project and led the collaboration. I.B. and A.C. performed the theoretical analysis for this work. H.B., T.He., T.Ho., S.I., A.F., T.K., A.S., L.K.U., M.G., and R.T. developed the tunable complex phase element and carried out the experimental implementation of unit-cell circuits. A.C., H.B., T.N., T.B., and I.B. wrote the manuscript. All authors discussed the results and commented on the manuscript.

## Competing interests
The authors declare no competing interests.
