## [Peer Review File · Nature Communications]

Hyperbolic Matter in Electrical Circuits with Tunable Complex PhasesREVIEWER COMMENTS

Reviewer #1 (Remarks to the Author):

In this manuscript, based on hyperbolic band theory proposed in PRB 105,125118 (2022), the authors introduced and experimentally realized hyperbolic matter by using topoelectrical circuit networks relying on a complex-phase circuit element. As an example of topologically nontrivial hyperbolic matter, they defined hyperbolic graphene as the collection of Bloch waves on the $\{10, 5\}$ lattice, and experimentally realized the unit-cell circuit for hyperbolic graphene in topoelectrics with four tunable complex-phase elements. They also theoretically demonstrated that hyperbolic graphene can exhibit characteristic gap opening in the Floquet regime.

Note that the experimental implementation of hyperbolic lattices has been reported in circuit quantum electrodynamics [ref.6 Nature 571, 45 (2019)] and topoelectrical circuits [ref.7 Nat. Commun. 13, 4373 (2022); ref.8 Nat. Commun. 13, 2937 (2022)]. In hyperbolic lattices, the ratio of bulk over boundary sites is of order unity no matter how large the graph, so that bulk observables on flakes always receive substantial contributions from the boundary sites. In these recent experiments, hyperbolic lattices were realized as finite planar graphs (flakes), consisting of bulk sites and boundary sites, and thus such experiments are not suited to probe the intrinsic bulk properties of hyperbolic lattices. Compared with the recent experimental implementation of hyperbolic lattices (refs. 6-8), a key advance of the present manuscript is that an experimental simulation of an imagined infinite hyperbolic lattice are presented in topoelectrical circuits with tunable complex-phase elements. Based on this technique, the pure bulk contribution to the nature of hyperbolic lattices is obtained. In my opinion, this is an important and exciting work which has significant impact on the field of topological hyperbolic lattices. This work paves the way for realizing more complex forms of hyperbolic matter and future experimental simulation of topological hyperbolic lattices. Therefore, I recommend publication in Nature Communications.

Reviewer #2 (Remarks to the Author):

The authors propose theoretically and realize experimentally the hyperbolic matter by electric circuits. They construct bulk hyperbolic matters by applying $U(1)$ hyperbolic band theory in a unit cell, where the influence of hyperbolic boundary sites can be removed. In addition, they numerically demonstrate the effectiveness of such a method by comparing the DOS of hyperbolic matters calculated by $U(1)$ band theory of a unit cell and the direct diagonalization of finite lattice with open boundaries. A good consistence appears, indicating that the effect of higher-dimensional representations of non-abelian translation group can be ignored. Furthermore, they propose the hyperbolic graphene, and experimentally realize one unit of such a structure, where the home-made tunable complex-phase circuit elements are used. Finally, the authors also argue that by introducing time modulation, non-trivial band gaps of hyperbolic graphene can be produced.

In my opinion, this is a very interesting work on hyperbolic matters. It can promote future investigations on the construction of novel hyperbolic matters with exotic properties. And, the tunable complex-phase element developed here can also play a useful role for future experimental simulations of novel topological states in circuits. Before I can recommend publication, I would like the authors to address the following issues/questions.

- 1.The $U(1)$ Bloch Hamiltonian of 2D hyperbolic matters are identical to that of a Euclidean model in higher-dimensions. The experimentally constructed 'hyperbolic graphene' is more like a 4D Euclidean model. Because, $U(1)$ Bloch theory is only complete for abelian translation group. It may be useful to construct abelian hyperbolic clusters with periodic conditions (several units of hyperbolic graphene with periodic

boundaries, discussed in Ref. 21), where the hyperbolic band theory with discrete k -vectors exist. Such a finite hyperbolic matter is more convincing to simulate curved spaces. Some numerical (or experimental) results related to hyperbolic graphene with finite units are very useful to illustrate the novel physics of hyperbolic graphene.

2. One of the interesting features of non-trivial Dirac semimetals is the existence of exotic boundary effects. It is useful to discuss the boundary effects of finite hyperbolic graphene with open boundaries.

3. Experimental results in Fig. 4d show a good consistence with theory except for that near gamma point. What does the little deviation result from?

4. The novel phenomena originated from Dirac points in Euclidean space (Zitterbewegung and Klein tunneling) have been observed in electric circuits [Communications Physics 4, 250 (2021)]. Do these novel phenomena still exist at hyperbolic Dirac points? It is very interesting to add some comments and expectations on the hyperbolic Dirac physics.

Reviewer #3 (Remarks to the Author):

This paper develops a tunable complex phase circuit element to emulate infinite hyperbolic lattice using unit-cell circuits. The validity of the emulation in this paper is based on Bloch-wave assumption which is the 1-dimensional representation of the nonabelian hyperbolic translation group. Their method could also be generalized to higher dimensional representation band theory. In this paper, hyperbolic lattice $\{7,3\}$, $\{8,3\}$, $\{8,4\}$, $\{10,3\}$ and $\{10,5\}$ are used to numerically verify the validity of Bloch-wave assumption and the band structure of hyperbolic graphene are measured using the newly developed technique. Instead of isolated Dirac points, the Dirac points in hyperbolic graphene forms a nodal surface in the 4-D momentum space and it has nontrivial topology. The new technique also opens the door of emulating Floquet system by periodically tuning complex phase circuit element.

Overall, the paper is well written, and the tunable complex phase circuit element opens many new possibilities such as emulate bulk states in other types of systems and emulate the Floquet states. The main novelty of this new experiment lies in its capability to emulate effectively infinite hyperbolic lattices without the extensive boundary. Before this work, hyperbolic lattices have been only realized as finite planar graphs (flakes), making it difficult to observe bulk properties. I recommend publishing this manuscript on Nature Communications.

In addition, I have the following comments/questions.

1. Since the hyperbolic translation group is nonabelian, equation (1) is better written as $\psi_{\mathbf{k}}(T_{\mu}^{-1} z_i) = e^{i\mathbf{k} \cdot \mu} \psi_{\mathbf{k}}(z_i)$, which makes the relative transformation of coordinate and wave function and the action of multiple translations more natural.

Although this makes no difference for 1-D representation, it matters for high dimensional representation. This notation has been used in Refs [21,22].

2. For the $\{7,3\}$ lattice, the author chose a flake with 847 sites. However, the Bravais lattice of $\{7,3\}$ is $\{14,7\}$, which has 56 sites in each unit cell and 847 sites corresponding to roughly 15 unit cells. Is it possible to use a larger flake to better examine the problem? 7 fold rotational symmetry may be useful to reduce the size of matrix by a factor of 7.

3. For the $\{8,3\}$ lattice, does the DOS of every regular map agree with the DOS from hyperbolic band theory?

4. The authors calculated the nontrivial Berry phase for the loop surrounding the Dirac point in the normal plane. Can the authors comment on the bulk-edge correspondence, or other observable effects, of this Berry phase in the context of hyperbolic lattices?

5. The author proposed that their newly developed method can be used to emulate Floquet dynamics by periodically tuning complex phase elements. It would be

interesting to show the opening of a gap at some Dirac point on the nodal surface. If this experiment is hard, can the author comment on why?

Point-by-point response to reviewers' comments:

In the following, we use blue color for our response to the Referees' comments, which will be displayed in black.

Reviewer #1

In this manuscript, based on hyperbolic band theory proposed in PRB 105,125118 (2022), the authors introduced and experimentally realized hyperbolic matter by using topoelectrical circuit networks relying on a complex-phase circuit element. As an example of topologically nontrivial hyperbolic matter, they defined hyperbolic graphene as the collection of Bloch waves on the $\{10, 5\}$ lattice, and experimentally realized the unit-cell circuit for hyperbolic graphene in topoelectrics with four tunable complex-phase elements. They also theoretically demonstrated that hyperbolic graphene can exhibit characteristic gap opening in the Floquet regime.

Note that the experimental implementation of hyperbolic lattices has been reported in circuit quantum electrodynamics [ref.6 Nature 571, 45 (2019)] and topoelectrical circuits [ref.7 Nat. Commun. 13, 4373 (2022); ref.8 Nat. Commun. 13, 2937 (2022)]. In hyperbolic lattices, the ratio of bulk over boundary sites is of order unity no matter how large the graph, so that bulk observables on flakes always receive substantial contributions from the boundary sites. In these recent experiments, hyperbolic lattices were realized as finite planar graphs (flakes), consisting of bulk sites and boundary sites, and thus such experiments are not suited to probe the intrinsic bulk properties of hyperbolic lattices. Compared with the recent experimental implementation of hyperbolic lattices (refs. 6-8), a key advance of the present manuscript is that an experimental simulation of an imagined infinite hyperbolic lattice are presented in topoelectrical circuits with tunable complex-phase elements.

Based on this technique, the pure bulk contribution to the nature of hyperbolic lattices is obtained. In my opinion, this is an important and exciting work which has significant impact on the field of topological hyperbolic lattices. This work paves the way for realizing more complex forms of hyperbolic matter and future experimental simulation of topological hyperbolic lattices. Therefore, I recommend publication in Nature Communications.

We thank the Referee for the encouraging review of our manuscript and the positive feedback.

Reviewer #2:

The authors propose theoretically and realize experimentally the hyperbolic matter by electric circuits. They construct bulk hyperbolic matters by applying $U(1)$ hyperbolic band theory in a unit cell, where the influence of hyperbolic boundary sites can be removed. In addition, they numerically demonstrate the effectiveness of such a method by comparing

the DOS of hyperbolic matters calculated by U(1) band theory of a unit cell and the direct diagonalization of finite lattice with open boundaries. A good consistence appears, indicating that the effect of higher-dimensional representations of non-abelian translation group can be ignored. Furthermore, they propose the hyperbolic graphene, and experimentally realize one unit of such a structure, where the home-made tunable complex-phase circuit elements are used. Finally, the authors also argue that by introducing time modulation, non-trivial band gaps of hyperbolic graphene can be produced.

In my opinion, this is a very interesting work on hyperbolic matters. It can promote future investigations on the construction of novel hyperbolic matters with exotic properties. And, the tunable complex-phase element developed here can also play a useful role for future experimental simulations of novel topological states in circuits. Before I can recommend publication, I would like the authors to address the following issues/questions.

We thank the Referee for the detailed and insightful review of our manuscript. We considered and evaluated all suggestions and implemented them as new sections, figures, or comments in the revised version of our manuscript. In the following we respond to each point individually.

1. The U(1) Bloch Hamiltonian of 2D hyperbolic matters are identical to that of a Euclidean model in higher-dimensions. The experimentally constructed 'hyperbolic graphene' is more like a 4D Euclidean model. Because, U(1) Bloch theory is only complete for abelian translation group. It may be useful to construct abelian hyperbolic clusters with periodic conditions (several units of hyperbolic graphene with periodic boundaries, discussed in Ref. 21), where the hyperbolic band theory with discrete k-vectors exist. Such a finite hyperbolic matter is more convincing to simulate curved spaces. Some numerical (or experimental) results related to hyperbolic graphene with finite units are very useful to illustrate the novel physics of hyperbolic graphene.

The Referee points out a very important aspect of hyperbolic graphs with periodic boundary conditions. The existence of Abelian hyperbolic clusters with periodic boundary conditions has been verified in Ref. 21. On an Abelian cluster, hyperbolic band theory is exact and reproduces all eigenstates of the tight-binding Hamiltonian. As of yet, however, no constructive algorithm for generating large Abelian (or non-Abelian) clusters exists. In on-going work, we are implementing ideas to construct large clusters for various hyperbolic lattices, including {10,5}, but this very nontrivial application of computational group theory is unfinished and not within the scope of the present work. As such, the regular maps (taken from the literature) used in the current analysis are the state-of-the-art of our knowledge about hyperbolic clusters.

However, as a first step towards a numerical investigation of Abelian clusters in the present manuscript, we constructed higher-dimensional Euclidean lattices by patching together unit-cells of hyperbolic {p,q} lattices in a 2g-dimensional Euclidean lattice. Such graphs exactly reproduce hyperbolic band theory by construction. Remarkably, for the {8,3} lattice, we can show that the graphs obtained are identical to certain regular maps. For the other choices of {p,q} that we studied, we could not make this identification, mostly because regular maps of sufficiently large size are not available in the literature. For the lattices {7,3}, {8,3}, {10,3}, {8,4}, {10,5} we constructed large clusters with N sites and

verified that the N eigenvalues of the adjacency matrix [labeled ϵ_i with $i=1,\dots,N$] exactly agree with the predictions of hyperbolic band theory [labeled $\epsilon(k_i)$] from a simple quantization condition applied to the momentum components k_i in the $2g$ -dimensional Brillouin zone. The following plot confirms this agreement, with the eigenvalues ϵ_i and $\epsilon(k_i)$ colored blue and red, respectively:

In the revised version of the manuscript, we have included a comprehensive and detailed discussion of the construction of such Euclidean clusters, together with the corresponding comparison to hyperbolic band theory in section S III of the supplemental material, which now features three justifications for the validity of hyperbolic band theory. We added a comment on Abelian clusters and reference to the Supplementary Material on page 3, right column, of the main text.

2. One of the interesting features of non-trivial Dirac semimetals is the existence of exotic boundary effects. It is useful to discuss the boundary effects of finite hyperbolic graphene with open boundaries.

We thank the Referee for the excellent suggestion to include a discussion of the bulk-boundary correspondence for hyperbolic graphene as a topological semimetal in the manuscript.

In its idealized version of fermions hopping on a honeycomb lattice, semimetallic graphene is a topological semimetal with zero-energy boundary states [arXiv:1301.0330]. The reason for this is that for any one-dimensional cut through the two-dimensional Brillouin zone (avoiding a Dirac point), the Bloch wave Hamiltonian realizes a one-dimensional topological insulator in class AIII with protected boundary states in position space. We confirm this behavior in a numerical diagonalization of a {6,3} flake: while the bulk DOS is small near zero energy, the edge DOS (defined as the difference between total DOS and bulk DOS) shows a pronounced peak at zero energy. Note that for this argument to work, the equality of dimension of position and momentum space is crucial.

The bulk topology of hyperbolic graphene is the four-dimensional analogue of graphene, as demonstrated by the π Berry phase around each Dirac node in the band-touching manifold. However, a similar theoretical construction of cuts in momentum space remains inconclusive, since position and momentum space have different dimensions. While first studies on the topological properties of hyperbolic lattices have appeared recently in Refs. [8,17,27,38], the interplay between position and momentum space invariants remains an

open problem.

For this reason, we addressed the presence of boundary states in hyperbolic graphene from an unbiased numerical point of view. By using a finite-sized $\{10,5\}$ flake with 7040 sites, we compare bulk DOS and edge DOS. We observe that there is no pronounced peak of edge states at zero energy. While some energy ranges are strongly populated with edge states, these regimes do not coincide with regions of small bulk DOS so that their topological interpretation is questionable. On the other hand, we cannot fully exclude the possibility that topological boundary modes are present (at least in the sector of Bloch-wave states), as they might be obscured by (i) the inherent inaccuracy in the separation of total DOS into bulk and edge contributions and (ii) the nonabelian eigenstates present in flakes but excluded by band theory. The former is particularly true for $\{10,5\}$ -flakes, which have an enormous fraction of boundary sites.

In the revised manuscript, we have included the discussion of the bulk-boundary correspondence for graphene and hyperbolic graphene on finite flakes in Sec. S V of the supplemental material. We added a comment in the bulk-boundary correspondence in graphene and hyperbolic graphene on page 5, left column of the main text, with a reference to the discussion in the Supplement. We further reference this important question at the end of the outlook.

3. Experimental results in Fig. 4d show a good consistence with theory except for that near gamma point. What does the little deviation result from?

The deviation in the plotted experimental data was caused by output limitations of the multipliers in the circuit. We measured another set of data with a lower input signal 0.5 V into the circuit (compared to originally 1 V). The multipliers perform better with the lower input signal, and the resulting experimental curve agrees nearly perfectly with theory (see figure below).

In the revised version of the manuscript, we have replaced Fig. 4d by the plot of the improved data. We accordingly removed in the caption of Fig. 4 the sentence: "The discrepancies around the Γ -point are likely caused by output limitations of the multipliers in the circuit."

In this context note also that the overall energy scale measured in the topoelectrical circuit is fixed by a global rescaling of the circuit Laplacian. We choose the rescaling such that it matches the theoretical model. For consistency, we relabelled the y-axes of Figs 4c and 4d by "admittance [$1/\Omega$]", which is the experimentally measured number.

4. The novel phenomena originated from Dirac points in Euclidean space (Zitterbewegung and Klein tunneling) have been observed in electric circuits [Communications Physics 4, 250 (2021)]. Do these novel phenomena still exist at hyperbolic Dirac points? It is very interesting to add some comments and expectations on the hyperbolic Dirac physics.

The Referee raises a very exciting and important question. The study of these two properties of Dirac particles known from Euclidean space in the context of hyperbolic graphene shows great potential to reveal new physics. We believe, however, that such an investigation would first require a much better understanding of the interplay between position and momentum space in hyperbolic lattices. As we have elaborated (and numerically shown) in the discussion of the bulk-boundary correspondence under 2., it appears that some features of Dirac particles in graphene do not immediately apply to hyperbolic graphene or similar systems, because they rely on position and momentum space having the same dimension. We feel that Klein tunneling and Zitterbewegung fall into this category.

For one, Klein tunneling as tunneling through a one-dimensional potential step requires a clear notion of what a "one-dimensional" motion is. In the excellent reference pointed out by the Referee, the one-dimensional chain specifies this direction. In the context of graphene, one considers a potential with a step in the x-direction and infinite potential in the y-direction. Klein tunneling then results from the Dirac equation with a spatially varying potential in one direction. However, while a potential step can be implemented in position

space on a hyperbolic flake, it would not correspond to one particular direction of momentum space in the Dirac equation, and vice versa. Hence it is not obvious how the Klein effect would emanate. In a similar way, the Zitterbewegung, while one expects the presence of positive and negative energy bands to yield the effect, is usually derived from an interplay of components of position- and momentum-operators in the Dirac equation, x_i and p_i , which requires an understanding of how their relative dimensions are to be matched.

We emphasize again that we believe that pursuing these two questions is extremely exciting, but rather material for one or more future investigations and publications. They may be imperative for identifying the relation between position and momentum space in hyperbolic matter in the future.

In the revised manuscript, we have added a comment/suggestion on studying Klein tunneling and Zitterbewegung, together with references, in the outlook session on page 6

Reviewer #3:

This paper develops a tunable complex phase circuit element to emulate infinite hyperbolic lattice using unit-cell circuits. The validity of the emulation in this paper is based on Bloch-wave assumption which is the 1-dimensional representation of the nonabelian hyperbolic translation group. Their method could also be generalized to higher dimensional representation band theory. In this paper, hyperbolic lattice $\{7,3\}$, $\{8,3\}$, $\{8,4\}$, $\{10,3\}$ and $\{10,5\}$ are used to numerically verify the validity of Bloch-wave assumption and the band structure of hyperbolic graphene are measured using the newly developed technique. Instead of isolated Dirac points, the Dirac points in hyperbolic graphene forms a nodal surface in the 4-D momentum space and it has nontrivial topology. The new technique also opens the door of emulating Floquet system by periodically tuning complex phase circuit element.

Overall, the paper is well written, and the tunable complex phase circuit element opens many new possibilities such as emulate bulk states in other types of systems and emulate the Floquet states. The main novelty of this new experiment lies in its capability to emulate effectively infinite hyperbolic lattices without the extensive boundary. Before this work, hyperbolic lattices have been only realized as finite planar graphs (flakes), making it difficult to observe bulk properties. I recommend publishing this manuscript on Nature Communications.

We thank the Referee for the thoughtful and inspiring review of our manuscript. We were able to integrate all suggestions into new sections, figures, or comments in the revised version of the manuscript. In the following, we respond to each point raised individually.

In addition, I have the following comments/questions.

1. Since the hyperbolic translation group is nonabelian, equation (1) is better written as $\psi_k (T_\mu^{-1} z_i) = e^{ik_\mu} \psi_k (z_i)$, which makes the relative transformation of coordinate and wave function and the action of multiple translations more natural. Although this makes no difference for 1-D representation, it matters for high dimensional

representation. This notation has been used in Refs [21,22].

We thank the Referee for this crucial suggestion and implemented the change in Eq. (1) of the revised version of our manuscript. Our notation is now consistent with Refs. [21,22].

2. For the $\{7,3\}$ lattice, the author chose a flake with 847 sites. However, the Bravais lattice of $\{7,3\}$ is $\{14,7\}$, which has 56 sites in each unit cell and 847 sites corresponding to roughly 15 unit cells. Is it possible to use a larger flake to better examine the problem? 7 fold rotational symmetry may be useful to reduce the size of matrix by a factor of 7.

The Referee points out an important test of the validity of our claims made. We were able to generate a 6-shell $\{7,3\}$ lattice with 2240 sites, 1008 of which are boundary sites. When plotted together with the existing data of Fig. 3(c) for 5 shells, the bulk-DOS of the 6-shell $\{7,3\}$ lattice (denoted by $\{7,3,6\}$ below) agrees perfectly with that of the 5-shell $\{7,3\}$ lattice (denoted $\{7,3,5\}$ below), although the numerical data are not exactly the same, see the figure shown below.

As discussed in our work, the remaining discrepancy between the bulk-DOS of flakes and the band-theory curve is either (i) due to the eigenstates which are higher-dimensional irreducible representations of the Fuchsian group (excluded from the band theory but present in flakes) or (ii) from a consistent boundary contribution to the bulk-DOS which does not diminish with increasing system size. The latter cannot be excluded due to the fact that the boundary-to-total ratio of a hyperbolic lattice converges to a sizeable constant.

In the revised manuscript, we included a discussion of the analysis of larger $\{7,3\}$ flakes in Sec. S III of the supplemental material.

3. For the $\{8,3\}$ lattice, does the DOS of every regular map agree with the DOS from hyperbolic band theory?

We thank the Referee for raising this question and pointing out that the averaging over regular maps was not sufficiently well-motivated. We average over several regular maps

as they tend to have (likely accidental) degeneracies and finite-sized gaps in the energy spectra that we do not expect to represent the behavior of the infinite lattice. The averaging eliminates these non-universal features. However, the agreement between hyperbolic band theory and each individual regular map is of comparable quality to the data shown here, as we show for every regular map of $\{8,3\}$ -type in the figure below:

We included a brief discussion of this relevant aspect in the revised manuscript in the caption of Fig. S4.

4. The authors calculated the nontrivial Berry phase for the loop surrounding the Dirac point in the normal plane. Can the authors comment on the bulk-edge correspondence, or other observable effects, of this Berry phase in the context of hyperbolic lattices?

The Referee addresses a crucial aspect of topological semimetals that deserves to be discussed in our exposition of hyperbolic graphene. We have implemented according changes in the revised manuscript in response to Reviewer #2's point 2., and would kindly refer the Referee to the response we provide there and the according changes made to the revised manuscript listed there.

5. The author proposed that their newly developed method can be used to emulate Floquet dynamics by periodically tuning complex phase elements. It would be interesting to show the opening of a gap at some Dirac point on the nodal surface. If this experiment is hard, can the author comment on why?

The Referee points out an excellent question that inspired us to include another analysis in our revised manuscript. Periodically driving the complex circuit element in experiment, while technically possible, is an involved task in practice as the modulation needs to be carefully fine-tuned to the desired frequency to yield the Floquet effect. The complex implementation is beyond the scope of the present work, which is a proof of principle of the functionality of the device. However, we analyzed the Referee's question numerically and computed the gap opening due to the Floquet drive along the continuous line in momentum space that is shown in Fig. 4d without Floquet drive. We obtain the following theoretical energy spectrum:

Note that for the non-driven system, as shown in Fig. 4d, there is an extended gapless region from A to B, and a band-crossing point at C. The line from A to B corresponds to a continuous line on the nodal surface that the Referee is referring to. Consistent with our analysis presented in the manuscript, the Floquet drive opens up a gap for all Dirac points.

In the revised version of the manuscript, we included a discussion of this gap-opening in Figure S7, which now has two panels instead of one.

List of further changes made to the revised manuscript

Besides the changes made to the revised manuscript that we described above, the following have been incorporated additional:

- Adhered to journal formatting requirements together with smaller stylistic changes throughout the manuscript.
- Added ORCID reference for author A Chen
- Changed labeling of y-axis in Fig 4c and d (see Referee #2, point 3).

- Added Refs. [23,24,44-47]
- Added journal reference for Ref. [38]
- Rewrote the second paragraph in Supplementary Sec S II, with a new definition of κ , and adjusted values of κ in Supplementary table S2. This parameter κ does not appear in the main text.

In the revised manuscript, all newly added parts are colored blue for better distinguishability.

Comment on data and code availability for review

The data and documented code required to interpret and reproduce the results of this manuscript are publicly available under <https://doi.org/10.5683/SP3/EG9931>.

REVIEWERS' COMMENTS

Reviewer #2 (Remarks to the Author):

In the revised manuscript, the authors provided very careful and complete responses to the comments and suggestions raised by the reviewers. The reviewer thanks the authors' efforts during the revision and happily suggests the acceptance of the manuscript.

Reviewer #3 (Remarks to the Author):

The authors have addressed all my concerns. I believe that this paper represent significant new progress for hyperbolic lattices, and recommend publication on Nature Communications.

Point-by-point response to reviewers' comments:

In the following, we use blue color for our response to the Referees' comments, which will be displayed in black.

Reviewer #2

In the revised manuscript, the authors provided very careful and complete responses to the comments and suggestions raised by the reviewers. The reviewer thanks the authors' efforts during the revision and happily suggests the acceptance of the manuscript.

We thank the Referee for the encouraging review of our manuscript and the positive feedback.

Reviewer #3

The authors have addressed all my concerns. I believe that this paper represent significant new progress for hyperbolic lattices, and recommend publication on Nature Communications.

We thank the Referee for the encouraging review of our manuscript and the positive feedback.